# Serum Uric Acid Is Associated with Insulin Resistance in Non-Diabetic Subjects

**DOI:** 10.3390/jcm14082621

**Published:** 2025-04-11

**Authors:** Janis Timsans, Jenni Kauppi, Vappu Rantalaiho, Anne Kerola, Kia Hakkarainen, Tiina Lehto, Hannu Kautiainen, Markku Kauppi

**Affiliations:** 1Department of Rheumatology, Päijät-Häme Central Hospital, Wellbeing Services County of Päijät-Häme, 15850 Lahti, Finland; anne.kerola@helsinki.fi (A.K.); markku.kauppi@paijatha.fi (M.K.); 2Faculty of Medicine and Health Technology, Tampere University, 33100 Tampere, Finland; vappu.rantalaiho@tuni.fi; 3Unit of Physiatry and Rehabilitation Medicine, Päijät-Häme Central Hospital, Wellbeing Services County of Päijät-Häme, 15850 Lahti, Finland; jenni.kauppi@paijatha.fi; 4Centre for Rheumatic Diseases, Tampere University Hospital, 33521 Tampere, Finland; 5Department of Medicine, Kanta-Häme Central Hospital, 13530 Hämeenlinna, Finland; 6Institute for Molecular Medicine Finland, Helsinki Institute of Life Science, University of Helsinki, 00014 Helsinki, Finland; 7Department of Nephrology, Päijät-Häme Central Hospital, Wellbeing Services County of Päijät-Häme, 15850 Lahti, Finland; kia.hakkarainen@paijatha.fi; 8Department of Clinical Chemistry, Fimlab Laboratoriot Oy, 15140 Lahti, Finland; tiina.lehto@fimlab.fi; 9Folkhälsan Research Center, 00250 Helsinki, Finland; hannu.kautiainen@medcare.fi; 10Primary Health Care Unit, Kuopio University Hospital, 70029 Kuopio, Finland; 11Clinicum, Faculty of Medicine, University of Helsinki, 00014 Helsinki, Finland

**Keywords:** hyperuricemia, uric acid, insulin resistance, HOMA-IR, metabolic hyperuricemia, renal hyperuricemia

## Abstract

**Background**: Glucose metabolism disorders are major contributors to morbidity and mortality. Elevated serum uric acid (SUA) is closely linked to the cardiometabolic consequences of glucose metabolism disorders, various other comorbidities, and mortality. In this study, we explore the relationship between SUA and fasting plasma glucose (FPG), insulin levels, and insulin resistance in an older Finnish adult cohort. **Methods**: We used data from the GOAL (**GO**od **A**geing in **L**ahti region) study—a prospective, population-based study of Finnish individuals aged 52–76 years. A total of 2322 non-diabetic subjects were included in the study. Data of SUA, FPG, and other laboratory parameters, comorbidities, lifestyle habits, and socioeconomic factors were collected. Subjects with SUA values of >410 μmol/L (≈6.9 mg/dL; 75th percentile) were regarded as hyperuricemic. We investigated the relationship between SUA and FPG, insulin levels, and insulin resistance [homeostatic model assessment of insulin resistance (HOMA-IR) ≥2.65]. **Results**: We found statistically significant sex-, age- and BMI-adjusted small to moderate relationships (Cohen’s standard for β values above 0.10 and 0.30, respectively) between SUA and FPG, insulin levels, and insulin resistance in the whole study population as well as in the female and male subgroups. The higher the SUA level, the higher the HOMA-IR [(adjusted β = 0.21 (95% CI: 0.17 to 0.25)], and it rises drastically if SUA is above 400 μmol/L (≈6.7 mg/dL). The probability of a subject having insulin resistance is related to SUA level. **Conclusions**: Hyperuricemia is associated with elevated FPG and insulin resistance, emphasizing the importance of addressing both conditions. Further research may explore hyperuricemia treatment’s role in preventing glucose metabolism disorders and their cardiometabolic consequences.

## 1. Introduction

Diabetes is a persistent metabolic condition marked by increased blood glucose levels, resulting in long-term damage to the heart, blood vessels, eyes, kidneys, and nerves. The global health burden of diabetes is high—it affects approximately 830 million individuals globally, and each year over 1.5 million deaths are directly linked to diabetes. The incidence and prevalence of diabetes have also been consistently rising over recent decades [1]. The most common type of diabetes—type 2 diabetes (T2DM)—is a preventable disease [2,3], but it requires early recognition of individuals at risk.

Prior to being diagnosed with diabetes, individuals commonly experience an extended period of prediabetes [4], characterized by impaired fasting glucose, impaired glucose tolerance, or both. People with T2DM and prediabetes exhibit various degrees of insulin resistance [5]—a condition involving elevated blood sugar levels and the body’s compensatory response of increased insulin production [6]. In the prediabetic stage, insulin resistance is the most powerful predictor of future development of T2DM [7]. Therefore, the identification of insulin resistance is important in order to implement early intervention measures in individuals at risk of developing T2DM. Insulin resistance also complicates the treatment of diabetes. In addition, in the past few years, accumulating findings have indicated a synergistic association between insulin resistance and its compensatory hyperinsulinemia in the onset and advancement of specific cancer types. Insulin resistance is also considered a significant contributing factor in the development of cardiovascular and cerebrovascular diseases, fatty liver disease, polycystic ovary syndrome, postadolescent acne and gastro-esophageal reflux disease [8]. Insulin resistance is also an additional risk factor in the pathogenesis of cardiovascular disease in those who already have T2DM [9].

The euglycemic hyperinsulinemic clamp, introduced by DeFronzo et al. in 1979 [10], stands as the gold standard for assessing insulin resistance [8,11,12]. Widespread clinical application of it is limited due to its complexity and constraints [8,11]. Multiple surrogate indices have been developed and validated, utilizing insulin and glucose levels measured during fasting or in response to glucose challenges. The most widely used index for assessing insulin sensitivity is the homeostatic model assessment of insulin resistance (HOMA-IR) [13].

Hyperuricemia is an important health risk with rising prevalence [14]. Previous studies have shown a high prevalence of diabetes in persons with hyperuricemia and/or gout—the combined prevalence in persons with either condition reported in a recently published meta-analysis of 38 studies was 19.1% [15]. Additionally, the prevalence of hyperuricemia in diabetic individuals is high, ranging between 25.3 and 33.8% [16,17,18,19,20,21]. In our previous study, we found a statistically significant association of higher serum uric acid (SUA) levels and fasting glucose in an elderly Finnish population; there was also a statistically significant association between higher SUA and diagnosis of diabetes in women [22]. The available data regarding the association between elevated SUA and insulin resistance are, however, limited. It has been shown that hyperuricemia is an independent risk factor for insulin resistance in healthy young individuals [23]. In a few studies, it was observed that elevated SUA levels precede the onset of insulin resistance, underscoring a potential connection between hyperuricemia and insulin resistance [24,25,26]. The cause–effect relationships between insulin resistance and hyperuricemia remain unclear. In some studies, insulin resistance has been shown as a potential causal factor for hyperuricemia [27,28].

In this study we investigated the relationship between fasting plasma glucose (FPG), insulin levels, insulin resistance, and SUA levels in a non-diabetic older Finnish population.

The preliminary results of this study were presented as an abstract at the 68th Annual Scientific Meeting of the Japan College of Rheumatology [29] and at ACR Convergence 2024 [30].

## 2. Materials and Methods

### 2.1. Study Population

We utilized data from the GOAL (Good Ageing in Lahti region) study—a prospective, population-based study focused on the elderly residing in the catchment area of the Päijät-Häme central hospital (located in Lahti, Finland). The recruitment period for this study began on 1 February 2002, and ended on 31 December 2002. The study included individuals from three age cohorts (52–76 years): those born in 1926–1930, 1936–1940, and 1946–1950. Baseline visits occurred in 2002, with follow-up visits in 2005, 2008, and 2012. Mortality data until the end of 2018 are accessible. Of the 4272 subjects invited, 2815 (66%) responded. Data collected from each study subject included the SUA level and other blood parameters (creatinine, cystatin C, blood glucose and insulin, total cholesterol, low-density lipoprotein cholesterol, high-density lipoprotein cholesterol, triglycerides, C reactive protein (CRP), high sensitivity C reactive protein (hsCRP), and 25-hydroxyvitamin D), socioeconomic status, psychosocial background, education, income, lifestyle habits (smoking, alcohol consumption, and exercise), previously diagnosed medical conditions (hypertension, diabetes, coronary heart disease, stroke, and cancer), documentation of the use of medication, and hospitalization data for the 12 months preceding the baseline visit. The blood pressure of the study subjects was measured at baseline three times and the average was documented. The height and weight of the study subjects were measured, and the body mass index (BMI) calculated. The waist circumference was measured at a level midway between the lowest rib and the iliac crest.

For the purposes of this study, the data were accessed on 22 May 2023. The authors of this study did not have access to information that could identify individual participants.

To assess insulin resistance, we used the HOMA-IR index [31], (I0 mU/mL  ×  G0 mg/dL)/405, where I0 is fasting plasma insulin and G0 is fasting plasma glucose.

In our study, we present persons with SUA > 410 μmol/L (≈6.9 mg/dL; 75th percentile) as clearly hyperuricemic, and persons with a HOMA-IR ≥ 2.65 (threshold used in several other studies [32,33]) as having insulin resistance. We use a percentile-based cut-off value of SUA to define hyperuricemia, because there is no international consensus on the SUA cut-off for defining hyperuricemia and the percentile-based cut-off best describes hyperuricemia in the specific population of our study. Since it has been previously demonstrated that the etiology of hyperuricemia matters in regard to the risk of all-cause mortality and cardiovascular outcomes [34,35,36], we also did a subgroup analysis among clearly hyperuricemic individuals according to the etiology of hyperuricemia (renal vs. metabolic), where we defined hyperuricemic persons with an estimated glomerular filtration rate (eGFR) of ≤67 mL/min/1.73 m^2^ (25th percentile) as having renal hyperuricemia, and hyperuricemic persons with an eGFR of >67 mL/min/1.73 m^2^ as having metabolic hyperuricemia. The glomerular filtration rate was calculated using the Chronic Kidney Disease Epidemiology Collaboration (CKD-EPI) creatinine–cystatin C equation, which is an accurate marker for estimating renal function in elderly individuals [37].

### 2.2. Statistical Methods

Summary statistics were described using mean and standard deviation (SD), or numbers as percentages. Statistical evaluations between groups were analyzed using analysis of variance (ANOVA) or logistic models. Multivariate linear regression analysis was used to identify the relationship between SUA as a continuous variable and the insulin resistance measurements, with the standardized regression coefficient Beta (β). The Beta value is a measure of how strongly the predictor variable (SUA) influences the criterion (insulin resistance) variables. The Beta is measured in units of SD. Cohen’s standard for Beta values above 0.10, 0.30, and 0.50 represents small, moderate, and large relationships, respectively. A possible nonlinear relationship between SUA and the HOMA-IR was assessed by using 4-knot-restricted cubic spline general linear models (OLS regression and logistic analysis). The length of the distribution of knots was located at the 5th, 35th, 65th, and 95th percentiles. The statistical package Stata 18.0 StataCorp LP, College Station, TX, USA) was used for the analyses.

### 2.3. Ethics

The study followed the guidelines of the Declaration of Helsinki. The cohort study was approved in 2002 by the Ethics Committee of Päijät-Häme Central Hospital, which is located in the city of Lahti (ID number of approval: PHSP 2/2002/Q11 § 87). All participants gave their written informed consent prior to data collection. The use of the data gathered in the cohort study combined with registry data was approved by the HUS Regional Committee on Medical Research Ethics in 2019 (ID number of approval: HUS/1748/20219 § 124).

### 2.4. Patient and Public Involvement

Patients and/or the public were not involved in the design, conduct, reporting, or dissemination of plans of this research.

## 3. Results

From the 2815 persons who accepted the invitation to participate in the study, 2322 were non-diabetic and had baseline SUA level and main glucose metabolism values available. The flowchart detailing the selection of study subjects is shown in Appendix A. The mean age of the participants was 64 years (range 52–76 years).

The baseline characteristics of the study subjects by SUA levels (≥410 and <410 µmol/L) and HOMA-IR values (<2.65 and ≥2.65) are shown in Table 1.

Table 2 shows the relationships (Cohen’s standard for β values) between SUA and HOMA-IR values, FPG, insulin, and insulin to glucose ratio in an unadjusted model, sex-, age-, and BMI adjusted models, and a model adjusted for sex, age, BMI, mean arterial pressure (MAP), fasting plasma cholesterol level, leisure time physical activity, alcohol consumption, and smoking. We found statistically significant small to moderate relationships between hyperuricemia and all the aforementioned parameters in all models for both men and women, with the exception of the relationship between hyperuricemia and fasting plasma insulin in the unadjusted model in men.

The association of the SUA level and HOMA-IR value, as well as the association of the SUA level and risk of having a HOMA-IR value of ≥2.65, are demonstrated in Figure 1.

We further examined the variance in mean HOMA-IR values among individuals with normal SUA, those with renal hyperuricemia, and those with metabolic hyperuricemia, as illustrated in Figure 2. Hyperuricemic individuals had significantly higher HOMA-IR values compared to normouricemic individuals, with women having significantly higher HOMA-IR values than men. However, there was no statistically significant difference between individuals with metabolic hyperuricemia and those with renal hyperuricemia.

## 4. Discussion

In our study we detected a positive relationship between the SUA and HOMA-IR values, FPG, insulin, and insulin to glucose ratio in both women and men after adjusting for potential confounding factors. This finding is consistent with the previous studies done on the subject [38,39,40]. Data in non-diabetic subjects are, however, scarce, and to our knowledge, there are no similar studies conducted in the European populations.

Even though the association between elevated SUA and higher FPG, risk of diabetes and, according to some studies, insulin resistance seems to be certain, there is no conclusive evidence on the causational relationship between them [41]. It has been speculated that uric acid might be a causal factor in the development of insulin resistance, but reverse causality cannot be excluded. Fat accumulation in the liver is a common finding in people with insulin resistance, and it is caused by mitochondrial oxidative stress, in which intracellular uric acid plays an important role [42]. Elevated uric acid levels may induce the upregulation of monocyte chemoattractant protein-1 (MCP-1), a key mediator in adipocyte inflammation [43]. Uric acid was observed to directly impede insulin signaling, fostering insulin resistance, thus implicating it as a fundamental mechanism in the development of hepatic steatosis [44]. Moreover, uric acid has been shown to diminish the production of adiponectin, a hormone known for enhancing insulin action in fat cells and mitigating inflammatory responses. Consequently, hyperuricemia may contribute to adipocyte endocrine dysfunction by inciting low-level inflammatory reactions and insulin resistance [45].

It has also been hypothesized that insulin resistance could lead to higher SUA levels. Insulin resistance reduces renal excretion of uric acid on the proximal tubular of the kidney, leading to hyperuricemia [46]. In rats, insulin administration reduced urinary urate excretion while upregulating the expression of uric acid transporter 1 (URAT1) for urate reabsorption and downregulating ATP-binding cassette subfamily G member 2 (ABCG2) for urate secretion [47]. Glucose transporter 9 (GLUT9), also expressed in proximal renal tubular cells, acts as a high-capacity urate transporter. Insulin activates GLUT9a, facilitating urate reabsorption via the basolateral route in the proximal tubule [48].

Whether we can slow down the process of development of diabetes by reducing SUA remains a question yet to be answered. It has, however, already been shown that lowering uric acid with the xanthine oxidase inhibitor (XOI) allopurinol improved insulin resistance and systemic inflammation in asymptomatic hyperuricemia in a randomized controlled trial almost ten years ago [49]. Additionally, data from the Brisighella Heart Study suggest that lowering SUA with allopurinol treatment may have a positive effect on FPG levels in the general population [50]. Another XOI—febuxostat—has been shown to increase insulin sensitivity in primary gout patients [51]. A recent review on the management of diabetes with hyperuricemia concluded that the risk/benefit ratio of urate-lowering therapy (ULT) in hyperuricemic individuals with diabetes but without gout is unclear [52]. Data from diabetic rat models, however, clearly suggest that XOI offers benefits such as improving glucose tolerance and reducing insulin resistance [53], reducing albuminuria, and alleviating renal oxidative stress [54].

Previously, we found out that the etiology of hyperuricemia matters in regard to hyperuricemia-related all-cause and especially cardiovascular mortality risk—persons who develop hyperuricemia in the setting of normal renal function (metabolic hyperuricemia) are at greater risk than those who develop it as a result of renal disfunction (renal hyperuricemia) [34,35]. A similar finding has been found in a large Italian study, where a higher serum creatinine to serum urate ratio (indicative of hyperuricemia without significant renal disfunction) was found to be more hazardous in regard to cardiovascular events (acute myocardial infarction, angina pectoris, heart failure, stroke, transient ischemic attack, and hypertensive complications) [36]. In the light of these findings, we examined the variance in mean HOMA-IR values among individuals with normal SUA, those with renal hyperuricemia, and those with metabolic hyperuricemia separately. Unsurprisingly, the HOMA-IR levels were higher in the hyperuricemic individuals compared to the normouricemic individuals, yet there was no statistically significant difference between those with renal and metabolic types of hyperuricemia. We previously theorized that metabolic hyperuricemia poses a greater risk of all-cause and cardiovascular mortality compared to renal hyperuricemia because of overproduction of uric acid in metabolic hyperuricemia, leading to increased reactive oxygen species (ROS) generation by xanthine oxidases [55]. ROS-induced endothelial dysfunction contributes to cardiovascular morbidity and heightened mortality risk [56]. The lack of a statistically significant difference in insulin resistance between the renal and metabolic hyperuricemia groups may stem from the abundance of free fatty acids in individuals with insulin resistance [57]. These fatty acids likely contribute to oxidative stress, potentially overshadowing the additional reactive oxygen species (ROS) generated by xanthine oxidases during uric acid synthesis. Additionally, renal insufficiency itself is associated with insulin resistance [58], which might reduce the difference between the two types of hyperuricemia.

Our study has several significant strengths. It was carried out within a population-based framework, mirroring the demographic structure of a specific age group in a Nordic nation. With a participation rate of 66% among those invited, our enrollment ensured a robust representation of both sexes. Furthermore, we rigorously adjusted for numerous potential confounders, bolstering the reliability and precision of our results. As a marker of insulin resistance, we used HOMA-IR, which is the most widely used insulin resistance marker, with high reliability. To calculate the eGFR (estimated glomerular filtration rate) of the study subjects, we used the CKD-EPI creatinine–cystatin C equation, which is a very accurate marker for estimating renal function in elderly individuals [37].

An inherent limitation of our study is the potential underrepresentation of the most severely ill and disabled individuals. This could stem from their increased likelihood of reluctance or inability to participate in the study, thus affecting the generalizability of our findings.

For baseline comparisons, study subjects were categorized into four groups based on SUA level (cutoff: 410 µmol/L, 75th percentile) and HOMA-IR (cutoff: 2.65). The groups were unevenly distributed, with a significantly larger proportion in the SUA < 410 µmol/L and HOMA-IR < 2.65 group, making comparisons more challenging and posing a study limitation. However, this imbalance did not impact the results on association between SUA and insulin resistance, as SUA was analyzed as a continuous variable in relation to insulin resistance.

Even though in the models demonstrated in our study we have adjusted the results for potential confounders, there is always a probability of residual confounding factors and the inability to fully adjust for potential residual confounders, which may have influenced the observed associations and introduced bias into the results, which should be stated here as a limitation.

Another limitation is the inability to account for the use of ULT, diuretics, and other medications that may have influenced SUA levels. Unfortunately, we lack precise data on the type and duration of ULT use among those who received it. However, we do know that the number of ULT users in our study population was low.

Alcohol consumption is a significant factor influencing SUA levels, as ethanol increases uric acid production and reduces its renal excretion by altering kidney tubule function [59]. In our study, participants completed the Alcohol Use Disorders Identification Test (AUDIT) questionnaire, providing insight into their drinking habits. We adjusted the association between SUA and insulin resistance for AUDIT scores. However, a key limitation of the AUDIT questionnaire is its reliance on self-reported data, which may introduce inaccuracies and bias. Unfortunately, precise alcohol consumption data were unavailable, representing a limitation of our study.

## 5. Conclusions

Hyperuricemia is linked with elevated fasting plasma glucose (FPG) levels and insulin resistance, indicating a close interplay between uric acid levels and glucose metabolism. This highlights the significance of addressing both conditions in clinical management.

Furthermore, beyond its conventional role in managing gout, the use of urate-lowering treatment may hold promise as a preventive measure against the development of diabetes and its associated cardiometabolic complications. Lowering SUA in hyperuricemic patients with insulin resistance might have a beneficial effect on diabetes treatment outcomes. However, the precise mechanisms underlying the relationship between hyperuricemia, glucose metabolism disorders, and cardiovascular risk remain incompletely understood, warranting further investigation to elucidate the potential therapeutic implications of hyperuricemia treatment.

## Figures and Tables

**Figure 1 jcm-14-02621-f001:**
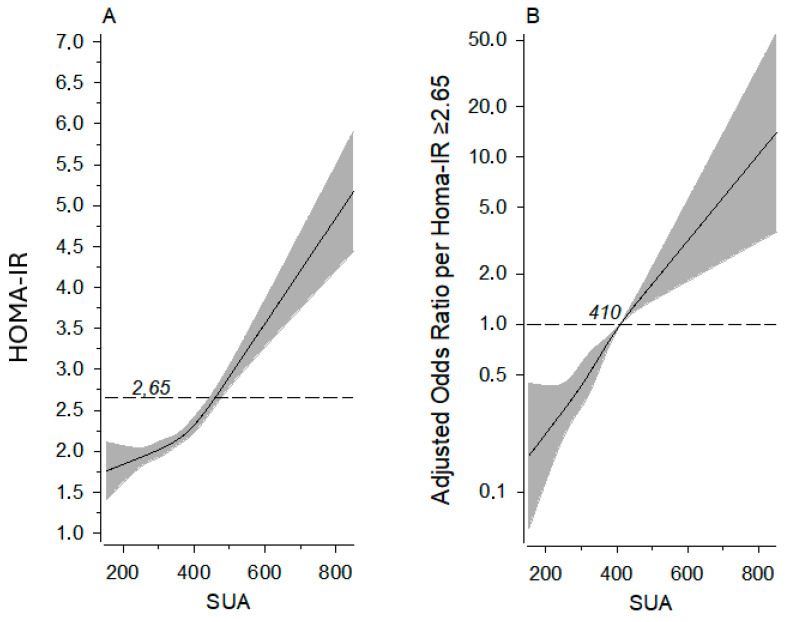
Panel (**A**): relationships of HOMA-IR as the function of the SUA level (dotted line shows HOMA-IR 2.65). Panel (**B**): probability of a person having HOMA-IR ≥ 2.65 by SUA level; dotted line shows odds ratio reference (OR 1.00). The curves were derived from four-knot-restricted cubic splines multivariate and logistic regression models. The models were adjusted for sex, age, BMI, MAP, fasting plasma cholesterol level, leisure time physical activity, alcohol consumption, and smoking. The gray area represents 95% confidence intervals. HOMA-IR, homeostatic model assessment of insulin resistance; SUA, serum uric acid.

**Figure 2 jcm-14-02621-f002:**
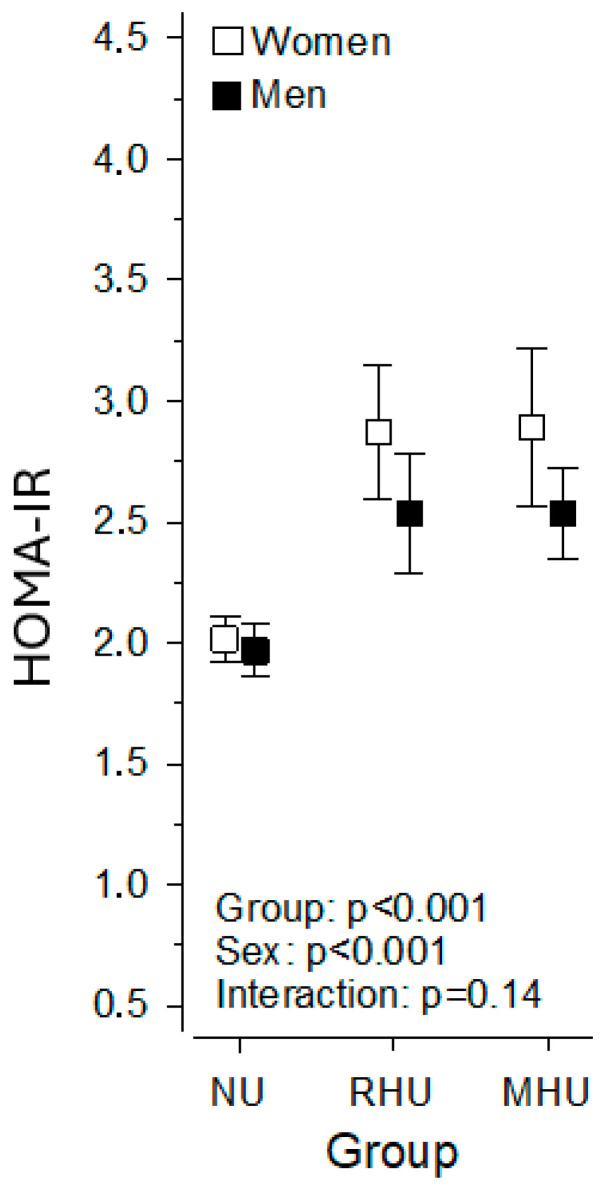
The mean HOMA-IR values among individuals with normal SUA, renal hyperuricemia, and metabolic hyperuricemia. HOMA-IR, homeostatic model assessment of insulin resistance; RHU, renal hyperuricemia; MHU, metabolic hyperuricemia; SUA, serum uric acid.

**Table 1 jcm-14-02621-t001:** Baseline characteristics of the study subjects by SUA (<410 μmol/L vs. ≥410 μmol/L) levels and HOMA-IR (<2.65 vs. ≥2.65) values.

	SUA < 410 µmol/L		SUA ≥ 410 µmol/L		*p*-Value		
	HOMA-IR <2.65	HOMA-IR ≥2.65	HOMA-IR <2.65	HOMA-IR ≥2.65	SUA	HOMA-IR	Interaction
	*n =* 1436	*n =* 343	*n =* 306	*n =* 237			
FPG, mean (SD), mmol/L	5.26 (0.47)	5.77 (0.54)	5.41 (0.50)	5.85 (0.50)	..	..	..
Fasting plasma insulin, mean (SD), µU/mL	6.41 (2.23)	15.22 (8.71)	7.26 (2.41)	16.68 (6.35)	..	..	..
Insulin/FPG ratio	1.2 (0.4)	2.7 (1.7)	1.3 (0.5)	2.9 (1.1)	..	..	..
HOMA-IR	1.50 (0.55)	3.87 (2.04)	1.75 (0.59)	4.35 (1.78)	..	..	..
SUA, µmol/L	313 (54)	341 (46)	457 (47)	478 (62)	..	..	..
Female, *n* (%)	560 (39)	128 (37)	221 (72)	147 (62)	<0.001	0.016	0.077
Age, mean (SD), years	63 (8)	66 (8)	64 (8)	65 (8)	0.23	<0.001	0.039
BMI, mean (SD), kg/m^2^	26.2 (3.7)	30.3 (4.5)	27.9 (3.8)	31.3 (4.9)	<0.001	<0.001	0.13
LTPA, *n* (%)					0.002	0.010	0.90
low	305 (21)	90 (26)	81 (26)	78 (33)			
moderate	623 (43)	148 (43)	136 (44)	100 (42)			
high	508 (35)	105 (31)	89 (29)	59 (25)			
BP, mmHg, mean (SD)							
systolic	143 (19)	150 (19)	146 (18)	153 (20)	0.006	<0.001	0.84
diastolic	85 (9)	88 (10)	86 (10)	90 (10)	0.005	<0.001	0.93
MAP	104 (11)	109 (11)	106 (11)	111 (12)	0.002	<0.001	0.87
Cholesterol, mean (SD), mmol/L	5.85 (1.03)	5.73 (1.05)	5.80 (1.16)	5.77 (1.10)	0.93	0.18	0.44
HDL-C, mean (SD), mmol/L	1.63 (0.45)	1.39 (0.35)	1.48 (0.40)	1.27 (0.33)	<0.001	<0.001	0.50
LDL-C, mean (SD), mmol/L	3.63 (0.91)	3.55 (0.95)	3.59 (1.03)	3.59 (0.97)	0.99	0.40	0.43
Triglycerides, mean (SD), mmol/L	1.30 (0.65)	1.73 (0.78)	1.68 (1.96)	2.03 (0.98)	<0.001	<0.001	0.40
eGFR, mean (SD), mL/min/1.73 m^2^	80.8 (14.1)	75.2 (14.3)	72.8 (15.5)	68.1 (15.9)	<0.001	<0.001	0.55
AUDIT score, mean (SD)	3.1 (2.3)	2.7 (2.4)	4.1 (2.7)	3.6 (2.7)	<0.001	0.001	0.88
Smoking, *n* (%)	40 (17)	51 (15)	54 (18)	35 (15)	0.84	0.23	0.80
Education years, mean (SD)	9.7 (3.3)	8.8 (2.9)	9.6 (3.4)	9.1 (3.5)	0.53	<0.001	0.18
Comorbidities, *n* (%)							
Antihypertensive	366 (25)	149 (43)	118 (39)	113 (48)	<0.001	<0.001	0.043
Cardiovascular disease	88 (6)	47 (14)	36 (12)	35 (15)	0.012	<0.001	0.049
Medication, *n* (%)							
Antihypertensive	294 (20)	128 (37)	99 (32)	109 (46)	<0.001	<0.001	0.24
Lipid lowering	176 (12)	62 (18)	59 (19)	45 (19)	0.029	0.11	0.081

SUA, serum uric acid; FPG, fasting plasma glucose; HOMA-IR, homeostatic model assessment of insulin resistance; BMI, body mass index; LTPA, leisure-time physical activity, graded as low (exercise less than once a week), moderate (one to two times a week), or high (at least three times a week); BP, blood pressure; MAP, mean arterial pressure; HDL-C, high-density lipoprotein cholesterol; LDL-C, low-density lipoprotein cholesterol; eGFR, estimated glomerular filtration rate, calculated using the CKD-EPI creatinine–cystatin C equation; AUDIT, Alcohol Use Disorders Identification Test.

**Table 2 jcm-14-02621-t002:** The relationships (Cohen’s standard for β values) between continuous serum uric acid (SUA) and HOMA-IR values, FPG, insulin, and insulin to glucose ratio. Model I: unadjusted. Model II: adjusted for sex (applies only to values of the whole study population), age, and BMI. Model III: adjusted for sex (applies only to values of the whole study population), age, BMI, MAP, fasting plasma cholesterol level, leisure time physical activity, alcohol consumption, and smoking.

	HOMA-IR Beta (95% CI)	FPG Beta (95% CI)	Fasting Plasma Insulin Beta (95% CI)	Insulin/FPG Ratio Beta (95% CI)
All				
Model I	0.33 (0.29 to 0.36)	0.27 (0.23 to 0.31)	0.30 (0.26 to 0.34)	0.26 (0.22 to 0.30)
Model II	0.21 (0.17 to 0.25)	0.13 (0.09 to 0.17)	0.19 (0.15 to 0.24)	0.15 (0.11 to 0.19)
Model III	0.21 (0.17 to 0.23)	0.11 (0.07 to 0.15)	0.20 (0.16 to 0.24)	0.18 (0.14 to 0.22)
Women				
Model I	0.35 (0.31 to 0.40)	0.26 (0.21 to 0.31)	0.33 (0.29 to 0.38)	0.30 (0.25 to 0.35)
Model II	0.18 (0.13 to 0.23)	0.15 (0.10 to 0.21)	0.17 (0.12 to 0.22)	0.15 (0.10 to 0.20)
Model III	0.19 (0.14 to 0.24)	0.13 (0.08 to 0.19)	0.18 (0.13 to 0.23)	0.16 (0.11 to 0.21)
Men				
Model I	0.32 (0.26 to 0.37)	0.16 (0.10 to 0.22)	0.11 (−0.07 to 0.29)	0.26 (0.21 to 0.32)
Model II	0.21 (0.16 to 0.27)	0.09 (0.03 to 0.16)	0.19 (0.14 to 0.25)	0.17 (0.11 to 0.23)
Model III	0.21 (0.15 to 0.26)	0.08 (0.02 to 0.14)	0.18 (0.14 to 0.25)	0.17 (0.12 to 0.23)

HOMA-IR, homeostatic model assessment of insulin resistance; FPG, fasting plasma glucose; SUA, serum uric acid; BMI, body mass index; MAP, mean arterial pressure.

## Data Availability

The data presented in this study are available on request from the corresponding author due to legal and ethical reasons.

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
