# Peer review of "Serum Uric Acid Is Associated with Insulin Resistance in Non-Diabetic Subjects"

_jcm, 2025, doi:10.3390/jcm14082621_

Round 1
Reviewer 1 Report
Comments and Suggestions for Authors
Dear Editor,
I carefully read the manuscript "Serum uric acid is linearly associated with insulin resistance in non-diabetic subjects".
My comments and suggestions for the authors are the following:
- In the abstract, the authors should include the sample size.
- Figure 2: "Gender" should be replaced by "Sex", as sex refers to a set of biological attributes in humans and animals. It is primarily associated with physical and physiological features including chromosomes, gene expression, hormone levels and function, and reproductive/sexual anatomy. In contrast, gender refers to the socially constructed roles, behaviours, expressions and identities of girls, women, boys, men, and gender diverse people.
- In the tables, the authors refer to "LDL" in place of "LDL-C" and to HDL" in place of "HDL-C". Please, revise.
- Table 1: English language needs to be carefully revised.
- The limitations of the study should be further and more deeply discussed in the manuscript.
- The authors should highly consider to refer to previous findings from the Brisighella Heart Study (e.g. doi: 10.1080/07853890.2016.1222451) in their manuscript.
Comments on the Quality of English LanguagePlease, see my comments for the Authors.
Author Response
Comment 1: In the abstract, the authors should include the sample size.
Response: The sample size has been added to the abstract in the revised manuscript.
Comment 2: Figure 2: "Gender" should be replaced by "Sex", as sex refers to a set of biological attributes in humans and animals. It is primarily associated with physical and physiological features including chromosomes, gene expression, hormone levels and function, and reproductive/sexual anatomy. In contrast, gender refers to the socially constructed roles, behaviours, expressions and identities of girls, women, boys, men, and gender diverse people.
Response: "Gender" has been replaced with "Sex" in Figure 2 in the revised manuscript.
Comment 3: In the tables, the authors refer to "LDL" in place of "LDL-C" and to HDL" in place of "HDL-C". Please, revise.
Response: "LDL" has been corrected to "LDL-C" and "HDL" to "HDL-C" in the revised manuscript.
Comment 4: Table 1: English language needs to be carefully revised.
Response: The English language has been carefully revised when preparing the revised version of the manuscript.
Comment 5: The limitations of the study should be further and more deeply discussed in the manuscript.
Response: The discussion of study limitations has been expanded in the revised manuscript (page 10, lines 308–335).
Comment 6: The authors should highly consider to refer to previous findings from the Brisighella Heart Study (e.g. doi: 10.1080/07853890.2016.1222451) in their manuscript.
Response: Thank you for the suggestion. We have now included a reference to the Brisighella Heart Study in the revised manuscript, as its findings are highly relevant to our discussion.
Reviewer 2 Report
Comments and Suggestions for Authors
- The term “diabetes mellitus” is now recognized solely as “diabetes” by the International Diabetes Federation (IDF).
- Be sure to define all abbreviations the first time they appear in the document (e.g. HOMA-IR, LDL, HDL, CKD-EPI, URAT1, ABCG2, etc.).
- Justify why individuals born in the years 1931-1935 and 1941-1945 are excluded.
- It is important to consider renal alterations in the included patients and the medications they take that can modify blood uric acid levels such as: xanthine oxidase inhibitors, uricosurics, diuretics and acetylsalicylic acid. If the data are not available, it should be taken into account that it may be a limitation to the study. This could be related to the results obtained for antihypertensive and lipid-lowering medications in individuals in the SUA <410 micromol/l and Homa-IR <2.65 groups.
- Authors should add 2 as a superscript in glomerular filtration rate units.
- There are 70 participants who are not included in the number of participants excluded from the study (493). Figure S1 mentions 423. Please correct or clarify this inconsistency.
- It is important to consider that the number of individuals in the four groups is disproportionate, which may lead to unsound conclusions due to this inequality, so it should be considered within the limitations of the study.
- The authors should discuss the importance of the AUDIT score, because ethanol acts as a diuretic and may be indirectly involved in the study results.
Author Response
Comment 1:The term “diabetes mellitus” is now recognized solely as “diabetes” by the International Diabetes Federation (IDF).
Response: Thank you for pointing this out. We have replaced “diabetes mellitus” with “diabetes” throughout the revised manuscript.
Comment 2: Be sure to define all abbreviations the first time they appear in the document (e.g. HOMA-IR, LDL, HDL, CKD-EPI, URAT1, ABCG2, etc.).
Response: All abbreviations are now defined at their first mention in the revised manuscript.
Comment 3: Justify why individuals born in the years 1931-1935 and 1941-1945 are excluded.
Response: Thank you for your comment. The individuals born in the years 1931-1935 and 1941-1945 were not excluded from the study. The study cohort consisted of individuals born in 1926-1930, 1636-1940 and 1946-1950, since originally the project was designed to compare three age cohorts from different decades. No exclusions were made based on birth years by the authors of this study.
Comment 4: It is important to consider renal alterations in the included patients and the medications they take that can modify blood uric acid levels such as: xanthine oxidase inhibitors, uricosurics, diuretics and acetylsalicylic acid. If the data are not available, it should be taken into account that it may be a limitation to the study. This could be related to the results obtained for antihypertensive and lipid-lowering medications in individuals in the SUA <410 micromol/l and Homa-IR <2.65 groups.
Response: We agree that medication affecting serum uric acid levels is relevant. While some data were available, they were incomplete (lacking details on type, duration, and dosage), so we did not include them in the analysis. This limitation is now acknowledged in the revised manuscript (page 10, lines 324–327).
Comment 5: Authors should add 2 as a superscript in glomerular filtration rate units.
Response: This has been corrected in the revised manuscript.
Comment 6: There are 70 participants who are not included in the number of participants excluded from the study (493). Figure S1 mentions 423. Please correct or clarify this inconsistency.
Response: Thank you for identifying this discrepancy. We have corrected the exclusion numbers in Figure S1 in the revised manuscript.
Comment 7: It is important to consider that the number of individuals in the four groups is disproportionate, which may lead to unsound conclusions due to this inequality, so it should be considered within the limitations of the study.
Response: We acknowledge this limitation, and it is now addressed in the revised manuscript (page 10, lines 312–318).
Comment 8: The authors should discuss the importance of the AUDIT score, because ethanol acts as a diuretic and may be indirectly involved in the study results.
Response: We have discussed the impact of ethanol on serum uric acid levels and clarified the use of the AUDIT score in assessing alcohol consumption (page 10, lines 328–335).
Reviewer 3 Report
Comments and Suggestions for Authors
This is iterative work by the authors expanding on their previous studies showing an association between serum uric acid (SUA) and fasting blood glucose (FBG) by examining the association between FBG, Insulin levels and resistance, and SUA in elderly patients from Finland. While interesting, this work is limited by the limited analysis and incremental nature of the findings. However, the work appears to be performed correctly and the author's interpretations are supported by the results.
The paper title references a linear relationship, the methods discuss a linear regression model, however there is no discussion in the text about this mathmatical relationship. Either the text needs to describe this relationship, or the focus needs to move away from the linearity aspect of the findings.
Figure 1- the figure lack panel identifiers, and the dotted line is labeled in panel B but not in Panel A.
Author Response
Comment 1: The paper title references a linear relationship, the methods discuss a linear regression model, however there is no discussion in the text about this mathmatical relationship. Either the text needs to describe this relationship, or the focus needs to move away from the linearity aspect of the findings.
Response: Thank you for your comment. We have revised the manuscript to de-emphasize linearity, removing the term “linear” from the title and abstract.
Comment 2: Figure 1- the figure lack panel identifiers, and the dotted line is labeled in panel B but not in Panel A.
Response: Thank you for your comment. We have added panel identifiers and labeled the dotted line in Panel A of Figure 1 in the revised manuscript.
Round 2
Reviewer 1 Report
Comments and Suggestions for Authors
Dear Editor,
I carefully read the revised version of the manuscript, that is significantly improved compared to the original version. I recommend its publication in the Journal.